# Determining the Elastic Constants of Isotropic Materials by Measuring the Phase Velocities of the A_0_ and S_0_ Modes of Lamb Waves

**DOI:** 10.3390/s23156678

**Published:** 2023-07-26

**Authors:** Olgirdas Tumšys, Liudas Mažeika

**Affiliations:** Ultrasound Research Institute, Kaunas University of Technology, K. Baršausko St. 59, LT-51423 Kaunas, Lithuania; liudas.mazeika@ktu.lt

**Keywords:** ultrasonic guided waves, material elastic constants identification, inverse problem, phase velocity of the fundamental modes

## Abstract

In this study, a new method for determining the elastic constants of isotropic plates using Lamb wave fundamental modes is presented. This method solves the inverse problem, where the elastic constants (Young’s modulus and Poisson’s ratio) of the plate were estimated by measuring the phase velocities of the Lamb wave using the Rayleigh–Lamb equations to find the solution and determining the phase velocities of the A_0_ and S_0_ modes using a new method. The suitability of the proposed method for determining the elastic constants was evaluated using simulated and experimental signals propagating on an aluminum plate. The theoretical modeling on the aluminum 7075-T6 plate shows that the proposed method allows the determination of the Poisson ratio with a relative error not exceeding 2% and Young’s modulus with a relative error not exceeding 0.5%. The experimental measurements of an aluminum plate of known thickness (2 mm) and density (2685 kg/m^3^) confirmed the suitability of the proposed method for the measurements of elastic constants. In the proposed method, the processing of ultrasonic signals can be performed in real-time, and the values of the elastic constants can be obtained immediately after scanning the required distance.

## 1. Introduction

Modern engineering constructions use new types of materials with non-standard and sometimes variable physical parameters. Therefore, determining the physical parameters of such materials is relevant to modern production processes and the robotization of these processes. Simple, inexpensive, fast-acting methods for determining these parameters are being intensively developed for production processes [1,2,3].

One of the methods for solving this problem, which enables the determination of the thickness, density, and elastic constants of plate-type materials, is the use of Lamb waves [4]. The characteristics of the propagation of these waves are sensitive to changes in the parameters of the material under investigation. The advantage of these waves is their propagation over long distances on plates with minimal energy loss and low amplitude attenuation. However, the determination of material parameters by Lamb waves is made more complicated due to the multimodal propagation of these waves and the dispersion of the signals [5]. Nevertheless, after properly choosing the excitation frequencies of the Lamb waves and the appropriate propagating modes, the physical parameters of the investigated plate can be determined by evaluating the signal propagation characteristics.

Lamb waves are used in studies investigating both isotropic and anisotropic plate structures. In simpler isotropic and homogeneous plates, only longitudinal and shear waves are present, which are characterized by their phase velocities, *c*_L_ and *c*_T_. The Lamb waves generated on the plate are a combination of these waves. The elastic constants of the plate influence the phase velocities of these waves. By knowing the phase velocities of Lamb wave propagation, it was possible to estimate the elastic constants of the plate, i.e., an inverse problem was being solved. Recently, there have been many methods of solving this problem.

Several directions can be distinguished in terms of the solutions to the inverse problem. One such direction was the use of fundamental Lamb modes at low frequencies when the product of the wave number *k* and the plate thickness *d* was *kd* << 1. In this case, approximate analytical expressions are used to describe the interdependence of elastic constants and phase velocity [6]. In [7], using laser-generated Lamb waves and the wavelet transform method, the researchers evaluated the thickness and elastic properties of metal foils with a thickness of less than 40 µm via the contact method. Another method [8] using low-frequency multimode ultrasonic Lamb waves was developed to measure any of the four acoustic parameters of thin elastic layers—thickness, density, shear, and longitudinal wave velocity—with respect to the other three parameters. This method was applied to the parameter measurements of thin aluminum layers, the thickness of which ranged from 26 µm to 512 µm. Additionally, the work in [9] presents a method for measuring the thickness of thin metal sheets with thicknesses ranging from 25 to 200 µm using antisymmetric A_0_ Lamb mode propagation. The main problem with the presented work, in this case, was that these methods could only determine the parameters of extremely thin plates.

In measuring the parameters of thicker plates, methods involving comparing the results of theoretical modeling and experimental measurements and looking for the smallest difference between these results have become widespread. The phase velocity of a plate with a known density depends on Young’s modulus and Poisson’s ratio. Usually, these parameters are determined by comparing the theoretically predicted Lamb wave dispersion curves with the curves of experimental data [10,11,12]. One of the most common methods for calculating theoretical dispersion curves for Lamb waves is the semi-analytic finite element (SAFE) method. This algorithm calculates the dispersion curves of Lamb waves by solving the dispersion wave equations using standard eigenvalue problem procedures [5]. The inverse problem of determining the elastic constants of isotropic plates was solved in a number of ways. The researchers in [13] describe an inverse technique based on the simplex method to determine the elastic constants of isotropic plates. Lower-order symmetric and antisymmetric modes of Lamb waves propagating along the plate in the signal frequency range of 100–650 kHz were used in this study. The method for determining the distribution of Young’s modulus in an isotropic plate based on velocity reconstruction tomography was described [14]. Here, Young’s modulus was related to the dispersion relations of Lamb modes. In solving the nonlinear inversion problem, velocity mapping was performed using the full-waveform inversion (FWI) method. The research in [15] proposes an inverse method to estimate the elastic constants of the material of the rails in use. The phase velocities of the Lamb wave modes are measured experimentally, and the dispersion curves are derived from them via SAFE numerical simulations. The elastic constants of the rails are determined through an inversion procedure based on an improved genetic algorithm (GA). To increase the speed and accuracy of the inverse procedure, particle swarm optimization (PSO) was chosen [16]. Elastic constants are determined by finding the smallest absolute percentage error between experimental and SAFE-calculated wave numbers. Experimental wavenumbers are calculated using the pencil decomposition method of the matrix (MPDM). In addition, the researchers in [17] propose an efficient modeling method-based identification method of material constants combining spatial Fourier transform and multiple signal classification (MUSIC) techniques. Only Young’s modulus was determined by this method. The obtained results show that Young’s modulus of the investigated aluminum plate was in very good agreement with that obtained using the traditional static testing of materials. Moreover, ref. [18] described an optical technique for determining the elastic constants of a material using maps of surface displacements obtained via pulsed TV holography. The phase velocity of the longitudinal wave was measured using the pulse-echo method. The calculated Poisson’s ratio and Young’s modulus were obtained with an accuracy better than 3% and 4%, respectively.

Numerical modeling methods are attractive because they can determine material parameters in complex geometries or anisotropic structures. However, their main advantage is also their disadvantage—the preliminary parameters of the tested plate must be known. Alternatively, a mismatch between experimental and simulated results can be obtained, and the relevant model parameters can be adjusted to match the physics.

Another common method for solving the inverse problem was to use elastic formulas and Lamb wave propagation functions to describe the propagation of signals in plates. These formulas are used to plot the phase velocity or wavenumber dispersion curves of Lamb waves. The inversion procedure uses various methods to recover the dispersion curves from the experimental results, followed by a search for the minimum residual ratio between the theoretical and experimental curves. The authors of [19] present a new elastic isotropic plate parameter hybrid computational system for material identification (HCSMI). Experimental dispersion curves and an artificial neural network (ANN) were used to determine these parameters. This method reduced the “gap” between the approximate experimental wavenumber curves and the theoretical dispersion curves obtained via direct analysis. The research results obtained proved the high efficiency of the HCSMI system for the identification of aluminum plate parameters. The inverse method was proposed in [20], based on an improved particle swarm optimization (PSO) algorithm to determine plate thickness and elastic constants. Numerical simulations and experimental studies have confirmed that Young’s modulus, Poisson’s ratio, and plate thickness can be accurately obtained from measured zero-order modal Lamb wave dispersion curves. The inversion method based on genetic algorithms (GA) was developed in the work [21]. It was designed for the wavenumber extraction of a single Lamb wave signal and the characterization of the plate. The proposed method realizes the wavenumber extraction and calculation of the plate parameters, avoiding overlapping Lamb wave modes and a low signal-to-noise ratio (SNR). As shown in the analysis of the simulated and experimental signals, the deviation in the wavenumber’s determination was about 1%, and the deviation in the plate parameters’ estimation was about 6%. A genetic algorithm was also used [22]. This method uses higher-order modes excited by a linear transducer array, and the elastic constants are determined using a comparison of theoretical and experimental wave numbers.

After analyzing the presented articles, the work’s goal was to create a simple and effective method for measuring the elastic constants of isotropic plate materials. A contact method was selected for exciting and receiving Lamb waves by scanning a certain plate area. The separation of fundamental modes was used, and Rayleigh–Lamb equations and the phase velocities of fundamental modes determined using a new method were used to solve the inverse problem. The technique of proposed method has several distinct advantages: (l) Only low-frequency fundamental Lamb wave modes are excited in the isotropic plate; (2) it was sufficient to only know the thickness and density of the plate in advance; (3) there was no need to calculate theoretical curves and compare them with experimentally obtained phase velocities; and (4) the processing of ultrasonic signals was performed in real-time, and values of elastic constants are obtained immediately after scanning the required distance.

The proposed method was described in the following order: In Section 2 of the article, the theoretical basis for determining the elastic constants was established by analyzing the phase velocities of Lamb waves. In Section 3, the influence of parameters on the uncertainty of the estimation of elastic constants was determined using simulated signals. The verification of the proposed method using experimental signals from an aluminum plate was given in Section 4. Section 5 presents the conclusions and looks at future research perspectives.

## 2. Theoretical Analysis

### 2.1. Fundamentals of Lamb Wave Propagation

Propagation of guided Lamb waves in homogeneous and isotropic plates depends on the elastic properties of the material (Young’s modulus, *E*, and Poisson’s ratio, *ν*), density (*ρ*), thickness (*d*), and frequency (*f*) of the guided wave. The propagation characteristics of these waves, depending on the listed parameters, are described by the analytical Rayleigh–Lamb equations [4]:(1)$$\frac{tan\left(q\frac{d}{2}\right)}{tan\left(p\frac{d}{2}\right)}=-\frac{4k^{2}pq}{{\left(q^{2}-k^{2}\right)}^{2}}—\mathrm{for the symmetric}\left(S_{i}\right)\mathrm{modes},$$
(2)$$\frac{tan\left(q\frac{d}{2}\right)}{tan\left(p\frac{d}{2}\right)}=-\frac{{\left(q^{2}-k^{2}\right)}^{2}}{4k^{2}pq}—\mathrm{for the antisymmetric}\left(A_{i}\right)\mathrm{modes},$$
where *k* = *ω*/*c*_p_ is the wavenumber, *ω* = 2π*f* is the angular frequency, and *c*_p_ is the phase velocity of the Lamb waves modes.

The variables *p* and *q* are related by the expressions:(3)$$p^{2}={\left(\frac{\omega }{c_{L}}\right)}^{2}-k^{2},\;q^{2}={\left(\frac{\omega }{c_{T}}\right)}^{2}-k^{2},$$
where *c*_L_ and *c*_T_ are longitudinal and transverse wave velocities, respectively.

Any Lamb wave mode in a plate is a combination of longitudinal and transverse waves influenced by the elastic constants of the plate:(4)$$c_{L}=\sqrt{\frac{E\left(1-\nu \right)}{\rho \left(1+\nu \right)\left(1-2\nu \right)}},\;c_{T}=\sqrt{\frac{E}{2\rho \left(1+\nu \right)}}.$$

The above equations describe the dependence of the dispersion characteristics of Lamb waves on the constants of the material. Meanwhile, the main parameters describing Lamb wave propagation based on these formulas are the phase velocity *c*_p_ and the frequency *f*.

Different modes of Lamb waves respond differently to changes in the material’s properties. However, using these formulas causes problems with the appearance of higher modes at higher frequencies. Then, at a fixed frequency, several phase velocity values corresponding to several different modes are obtained. When solving the inverse problem based on the phase velocity of Lamb waves, an unambiguous dependence between the phase velocity and these parameters is necessary to determine the elastic constants of the material. Therefore, for the solution of the inverse problem, a low frequency was chosen, where only the fundamental modes—A_0_ and S_0_—exist.

### 2.2. Determination of Elastic Constants Based on the Phase Velocities

The presented analytical Equations (1)–(4) describe the dependences of the dispersion curves of Lamb waves on the material parameters: Young’s modulus (*E*), Poisson’s ratio (*ν*), density (*ρ*), and thickness (*d*). Alternative forms of the dispersion Equations (1) and (2) are better suited for numerical simulations [5]:(5)$$\left\{\begin{array}{c} {\Psi_{SYM}\left(E,\nu ,\rho ,f,d\right)=\left(k_{S}^{2}-q_{S}^{2}\right)}^{2}cos\left(p_{S}\frac{d}{2}\right)sin\left(q_{S}\frac{d}{2}\right)+4k^{2}pqsin\left(p_{S}\frac{d}{2}\right)cos\left(q_{S}\frac{d}{2}\right)=0\\ \Psi_{ASYM}\left(E,\nu ,\rho ,f,d\right)={\left(k_{A}^{2}-q_{A}^{2}\right)}^{2}sin\left(p_{A}\frac{d}{2}\right)cos\left(q_{A}\frac{d}{2}\right)+4k^{2}pqcos\left(p_{A}\frac{d}{2}\right)sin\left(q_{A}\frac{d}{2}\right)=0 \end{array}\right.$$
where *k*_A_ = *ω*/*c*_pA_ is the A_0_ mode wavenumber, *k*_S_ = *ω*/*c*_pS_ is the S_0_ mode wavenumber, and variables *p* and *q* with subscripts correspond to modes A_0_ and S_0_.

The numerically found roots of these functions $\Psi_{SYM}$ and $\Psi_{ASYM}$ are (*ω*, *c*_pA,_
*c*_pS_). They can be calculated once we know the material parameters.

When solving the inverse problem, we assume that we are experimentally measuring the phase velocities of the A_0_ and S_0_ modes of the Lamb waves in the material. After that, the unknown material parameters must be calculated. However, there is a problem in this case—only two equations and four unknowns exist.

In experimental measurements, the thickness *d* of the material can be determined using mechanical means or ultrasonic pulse-echo methods. The material density *ρ* was either known in advance or measured by other methods. Therefore, in the selected inverse problem, we chose to determine Young’s modulus *E* and Poisson’s ratio *ν*. To calculate Young’s modulus and Poisson’s ratio, Equation (4) were used based on the velocities of longitudinal *c*_L_ and transverse *c*_T_ waves:(6)$$E={2c}_{T}^{2}\rho \left(1+\nu \right),\;\nu =\frac{2-{\left(\frac{c_{L}}{c_{T}}\right)}^{2}}{2\left(1-{\left(\frac{c_{L}}{c_{T}}\right)}^{2}\right)}.$$

The unknown elastic constants *E* and *ν* are found by finding the minimum of the objective function. For this, we used the standard Matlab function *fminsearch*. *fminsearch* uses the Nelder–Mead simplex algorithm [23] and finds the minimum of a multivariate scalar function starting from an initial estimate.

### 2.3. Measurement of the A_0_ and S_0_ Modes’ Phase Velocities

A new and simple experimental algorithm for determining the group and phase velocity of the A_0_ mode [24] based on signal filtering and zero-crossing estimation was used for the experimental determination of the phase velocities of the A_0_ and S_0_ modes of Lamb waves. In this method, the zero-crossing instances closest to the peaks of the signal envelopes are captured, and phase and group velocities are calculated using these time instances.

Zero-crossing instances concentrated in the environment of the signal envelopes’ peaks are determined by filtering the signals with the filtering algorithm [25]. When filtering the signals with different filters, a concentration of zero-crossing instances was observed on the time axis in the environment of the signal envelope peak (Figure 1b).

The concentration of zero-crossing instances in the time axis was determined by the minimum time difference between these instances [24]:(7)$$t_{iM}^{0}=arg\{min(\sum_{i=1}^{N-1}\left(\underset{1<k<K}{\min}\left(t_{ik}^{0}-t_{(i+1)k}^{0}\right)\right))\},$$
where $t_{iM}^{0}$ represents the concentrated zero-crossing instances; M is the number of zero-crossing instances in the *i*-th filter; *k* = 1, 2, …, K is the number of zero-crossing instances; K is the total number of zero-crossing instances; and *i* = 1, 2, … N, N is the total number of filters.

After performing these calculations, we obtained that the distance dependence of zero-crossing instances in narrow ranges was linear, with jumps for the A_0_ mode and only linear for the S_0_ mode (Figure 2a). A line drawn between two jumps for the A_0_ mode and a straight line drawn for the S_0_ mode form cases of equal phases of the investigated signals.

Based on the zero-crossing instances of the same phase, the phase velocities of the A_0_ and S_0_ modes of the Lamb waves for each case of the filtered signal are calculated:(8)$$c_{pAi}=\frac{x_{i3}-x_{i2}}{t_{i6}^{0}-t_{i3}^{0}},$$
(9)$$c_{pSi}=\frac{x_{i4}-x_{i1}}{t_{i2}^{0}-t_{i1}^{0}},$$
where *c*_pA*i*_ and *c*_pS*i*_ are the phase velocities of the A_0_ and S_0_ modes for the *i*-th filter, respectively.

The values of the phase velocities calculated by the algorithm described above are shown in Figure 3. This figure shows the dispersion curves calculated for an aluminum plate from the analytical expressions (Equations (1) and (2)) (lines). Meanwhile, the values of the phase velocities calculated by the proposed method are shown by dots. A filter packet of five filters (N = 5) was chosen for the calculations, and the excitation frequency was *f*_ex_ = 300 kHz. A detailed algorithm for selecting all parameters was presented in [24].

Theoretical simulations using simulated signals [24] have shown that the proposed method allows the calculation of the phase velocity with a mean relative error of less than 0.7%.

## 3. Estimation of Elastic Constants Using Simulated Signals

### 3.1. Formation of Simulated Signals for Fundamental Modes

In order to evaluate the reliability of the proposed method, simulating signals propagating in an aluminum plate 7075-T6 have been formed. The main parameters of this plate are Young’s modulus *E* = 71.7 GPa, Poisson’s ratio *ν* = 0.33, and density *ρ* = 2710 kg/m^3^. The excitation signal *y*(*t*) for the Lamb waves was a three-period harmonic signal with a Gaussian envelope. Then, the Lamb wave signal *u*(*x*,*t*) excited in a plate of constant thickness *d* at the propagation distance *x* was described by the following equation:(10)$$u\left(x,t\right)=\frac{1}{2\pi }\int_{-\infty }^{\infty }FT\left(y(t)\right)e^{-j\omega \frac{x}{c_{p}}}e^{j\omega t}d\omega ,$$
where FT is the Fourier transform, *t* is the time, and *j* is the basic imaginary unit $j=\sqrt{-1}$.

Since the signal *u*(*x*,*t*) depends on the phase velocity, we can describe the signals at specific distances and thus form a B-scan image after assuming the dependence of the phase velocity change. Based on the dispersion curves of the A_0_ and S_0_ modes (Figure 3) calculated using analytical expressions (Equation (5)), we formed B-scan images at a fixed excitation frequency, *f*_ex_, shown in Figure 4. The amplitudes of all A_0_ and S_0_ mode signals *(u*_A_(*x_r_*,*t_p_*) and *u*_S_(*x_r_*,*t_p_*)) that have traveled a distance *x_r_* are normalized to the maximum amplitude of the first signal *(u*_A_(*x*_1_,*t_p_*) or *u*_S_(*x*_1_,*t_p_*)). In experimental studies, the amplitudes of the S_0_ mode signals are usually lower than the A_0_ mode signals. Therefore, for the normalization of the signal amplitudes, a coefficient for the correction L_S_ of the S_0_ mode signal amplitudes has been introduced:(11)$$L_{S}=\frac{max\left(u_{S}\left(x_{1},t_{p}\right)\right)}{max\left(u_{A}\left(x_{1},t_{p}\right)\right)},$$
where *u*_A_(*x_r_*,*t_p_*) and *u*_S_(*x_r_*,*t_p_*) are the A_0_ and S_0_ mode signals, respectively.

The presented B-scan modeling methodology was used in further studies to calculate the phase velocities of the A_0_ and S_0_ modes from these B-scan images.

### 3.2. Parameter Selection of the Proposed Algorithm

The phase velocity determination algorithm has certain limitations and requirements for parameter selection [24]. The overlap of the two modes limits the method’s applicability because it distorts the phase of the analyzed mode. A certain scan distance is required to determine the phase velocity, which is caused by the fixed A_0_ mode propagation jump. The choice of excitation frequency also influences the results of the phase velocity calculation. The listed problems were solved during theoretical studies.

The propagation of A_0_ and S_0_ mode signals depends on the phase velocities of these modes (Equation (10)), and these velocities differ significantly at low frequencies. Thus, these signals are separated in time as they move away from the excitation transducer. By applying the time window method, these signals can be extracted. As reference points for the formation of time windows, we chose the maxima of the A_0_ and S_0_ mode signal envelopes (Figure 5):(12)$$e_{A}(x_{r})=max\left(HT[u_{A}(x_{r},t_{p})]\right),\;e_{S}(x_{r})=max\left(HT[u_{S}(x_{r},t_{p})]\right),$$
where HT is the Hilbert transform.

The time interval between the envelopes’ maxima Δ*t*_g_, which enables the separation of the A_0_ and S_0_ mode signals, was calculated:(13)$$∆t_{g}=e_{A}\left(x_{R}\right)-e_{S}\left(x_{R}\right)=\frac{ps}{f_{ex}},$$
where *ps* is the number of periods of the excited signal and *x*_R_ is the distance at which this condition is satisfied.

Applying this time window produces separate A_0_- and S_0_-mode B-scan images (Figure 5).

The calculated distance *x*_K_ determines the location from which it was possible to separate the modes and calculate their phase velocities *c*_pA_ and *c*_pS_.

Another important parameter is the distance at which sudden changes (jumps) in the phase of propagation of A_0_ mode signals are recorded. These jumps occur when the half-periods of the signal “move” within the signal envelope as the distance, phase, and group velocities differ. The distance between these jumps Δ*x*_T_ depends directly on the period *T* defined by the phase *c*_pA_ and group *c*_gA_ velocities:(14)$$T=\left|t_{gA}-t_{pA}\right|=\left|\frac{∆x_{T}}{c_{gA}}-\frac{∆x_{T}}{c_{pA}}\right|=\frac{∆x_{T}\left|c_{pA}-c_{gA}\right|}{{c_{gA}c}_{pA}},$$
(15)$$∆x_{T}=\frac{T{c_{gA}c}_{pA}}{\left|c_{pA}-c_{gA}\right|}=\frac{\frac{1}{f_{ex}}{c_{gA}c}_{pA}}{\left|c_{pA}-c_{gA}\right|},$$

Three jumps are observed during one period (Figure 6b). The distance between the two jumps, Δ*x*_T_/2, defines the minimum distance that must be scanned to determine the phase velocity *c*_pA_ of the A_0_ mode.

Another important task was the choice of the frequency of excitation. Since we use only fundamental modes, it is obvious that we need to choose an excitation frequency lower than the first Cutoff frequency for the phase velocity (Figure 7a, Cutoff frequency 1). On the other hand, it would be best to choose frequencies that provide the greatest sensitivity to changes in the elastic constants of the analyzed plate. For sensitivity assessment, we calculated the relative deviations Δ*_m_* of the phase velocities from the reference elastic constant (Young’s modulus, *E*, or Poisson’s ratio, *ν*) values, changing these values in 20% increments and decrement directions:(16)$${∆}_{m}=\frac{c_{0n}-\left(c_{0n}-c_{\%n}\right)}{c_{0n}},$$
where *c*_0*n*_ represents the calculated reference values of the phase velocities at the fixed frequencies, and *c*_%*n*_ represents the phase velocities calculated at the fixed frequencies using ±20% changed elastic constants. Large deviations in the elastic constants were chosen to determine how much such deviations influence the deviations of the phase velocities from the reference values. The calculated relative deviations Δ*_m_* are shown in Figure 7a, and the parameters of the calculations are listed in Table 1. The marking ◊ shows in which mode the reference values of the phase velocities are calculated.

In order to evaluate the influence of the elastic constants on the changes in the fundamental modes, the total relative deviation Δ_Σ_ was calculated. This deviation was normalized in relation to its minimum value and presented in Figure 7b.
(17)$${∆}_{\sum }=\sum_{n=1}^{8}\frac{c_{0n}-\left|c_{0n}-c_{\%n}\right|}{c_{0n}},\;\delta_{\sum }=\frac{{∆}_{\sum }}{min\left({∆}_{\sum }\right)}.$$

In order to evaluate the nature of the variation in the total relative deviation for different materials, the relative deviation for aluminum 7075-T6 and steel AISI 4340 was calculated. The same tendencies of variation in the total relative deviation were obtained.

As we can see from Figure 7b, it was possible to determine the frequency (Cutoff frequency 2, *f·d* ≅ 750 kHz·mm) from which a higher total deviation value was obtained.

### 3.3. Evaluation of Uncertainties in Simulation Results

Solving the inverse problem in order to determine the elastic constants of the materials gives results with certain errors. The relative error used to evaluate the suitability of the proposed method was obtained by calculating the elastic constants by the proposed method and comparing them with the initial values of these constants:(18)$$\delta_{\nu }=100\%\cdot \frac{\left|\nu -\nu_{c}\right|}{\nu },\;\delta_{E}=100\%\cdot \frac{\left|E-E_{c}\right|}{E},$$
where *ν* and *E* are the Poisson’s ratio and Young’s modulus initial values, respectively; *ν*_c_ and *E*_c_ are calculated elastic constants.

The relative errors are calculated using different excitation frequencies *f*_ex_ up to the Cutoff frequency 2. The starting frequency was 100 kHz, and the frequency step was 50 kHz. At the same time, the ratio of the amplitudes of the A_0_ and S_0_ modes (coefficient L_S_) was changed. The obtained results of relative error calculations are shown in Figure 8.

The calculations performed showed that it is possible to distinguish a frequency area with smaller errors. This area is practically the same in Poisson’s ratio and Young’s modulus calculations. It covers a stretch of *f·d* = 300–650 kHz·mm. In this area, the relative error of Poisson’s coefficient calculation does not exceed 2%, and the relative error of Young’s modulus calculation does not exceed 0.5%. This conclusion should be based on the selection of fundamental mode excitation frequencies *f*_ex_ for the specific thickness *d* of the plate under investigation.

## 4. Experimental Verification

Quantification of the proposed method for estimating elastic constants was performed using experimental studies of Lamb wave propagation in a *d* = 2 mm thick aluminum plate of 1.2 × 1.2 m^2^ in size. The elastic constants of the aluminum plate are not known exactly. The density of the aluminum plate was determined by weighing it (*ρ* = 2685 kg/m^3^).

The experimental B-scan was obtained with two contact transducers, one stationary as the transmitter and the receiver moving straight from the transmitter. Contact-type transducers with hemispherical plastic tips and a 220 kHz resonant frequency are used. The frequency of the transducers was selected according to the criterion of a small relative error for the corresponding thickness of the plate (Figure 8). The excitation signal of the transmitter was a three-period burst with a Gaussian envelope. The position of the receiver was changed with a Standa 8MTF-75LS05 scanner (Standa Ltd., Vilnius, Lithuania). Scanner control, signal excitation, and registration were carried out using the ultrasonic measurement system “Ultralab,” developed at the Ultrasound Research Institute of the Kaunas University of Technology.

The B-scan image was formed while the receiving transducer moved at a distance of 60–200 mm with a step of 0.1 mm (Figure 9a). Figure 9b shows the received signal at a distance of 100 mm from the sending transmitter. Figure 9c shows the amplitude–frequency characteristic of this signal.

It was determined that the ratio of the amplitudes of the A_0_ and S_0_ modes in experimental measurements was about L_S_ = 0.05. Meanwhile, the amplitude–frequency response of the signals was wide and different for A_0_ and S_0_ modes. This was revealed by separating one mode from another. Filter packets with different center frequencies are selected for these modes according to the algorithm described in detail [24]. Five filter packets (N = 5) have been selected for both modes. The following filter parameters have been selected for A_0_ mode: resonant frequency of the center filter *f*_3_ = 226.8 kHz; bandwidths of the filters Δ*f* = 46.4 kHz; and distances between the filters *df* = 34.8 kHz. Parameters corresponding to S_0_ mode are: *f*_3_ = 382.1 kHz, Δ*f* = 75 kHz, and *df* = 56.2 kHz.

The 2D-FFT method [26] was chosen to evaluate the intermediate results of the proposed method (phase velocities of A_0_ and S_0_ modes). The results of the B-scan processing of individual A_0_ and S_0_ modes by the 2D-FFT method are shown in Figure 10 with color marking. The values (dots) of the phase velocities calculated by the proposed method based on the experimental data are also presented in those pictures.

As we can see, the phase velocities of A_0_ and S_0_ modes obtained by both methods are very close. Next, the elastic constants of aluminum were calculated based on the values of these velocities and obtained as follows: Young’s modulus *E* = 68.85 GPa, Poisson‘s ratio *ν* = 0.35. The obtained elastic constants are very close to the aluminum plate constants given by [27]: *d* = 2 mm, *ρ* = 2700 kg/m^3^, *E* = 70 GPa, *ν* = 0.35. Yong’s modulus differs only by 1.6%, and the Poisson ratios match.

## 5. Discussion and Conclusions

This paper presents a new and simple method for determining the elastic constants of isotropic plates using Lamb waves’ fundamental modes. The proposed method solves the inverse problem where the elastic constants (Young’s modulus and Poisson‘s ratio) of the plate are estimated by measuring the phase velocities of the Lamb wave. Rayleigh–Lamb equations and the phase velocities of fundamental modes (A_0_ and S_0_) determined by a new method are used to solve the inverse problem. Theoretical modeling on an aluminum 7075-T6 plate showed that the proposed method allows the Poisson’s ratio to be determined with a relative error not exceeding 2% and Young’s modulus to be determined with a relative error not exceeding 0.5%. In the theoretical simulation, the selection of the excitation frequencies of the fundamental modes for the specific thickness of the investigated plate was justified, and the minimum required scanning distance of that plate was defined. Experimental measurements on a 2 mm thick aluminum plate confirmed the suitability of the proposed method for elastic constant measurements.

However, this method has some limitations and unexplored applications. The overlap of the different modes affects the method’s applicability at close distances between the transducers. Some initial scanning distance was required. A certain scan distance was required to determine the phase velocity of the A_0_ mode, which was determined by the fixed propagation jump of the A_0_ mode. The application of this method to complex composite plates has also not yet been investigated.

However, after evaluating the limitations of this method, it was necessary to emphasize the advantages of this method as well. First, the method does not require prior knowledge of the Lamb wave phase velocities’ curves or the preliminary values of the elastic constants of the plate under investigation. Second, the processing of the received ultrasonic signals can be performed in real-time, and the values of the elastic constants can be obtained immediately after scanning the required distance. This enables the method to be used in automated systems for determining material parameters. Since the method focuses only on the processing of received signals, this methodology could be applied using non-contact excitation and the receiving of Lamb waves.

## Figures and Tables

**Figure 1 sensors-23-06678-f001:**
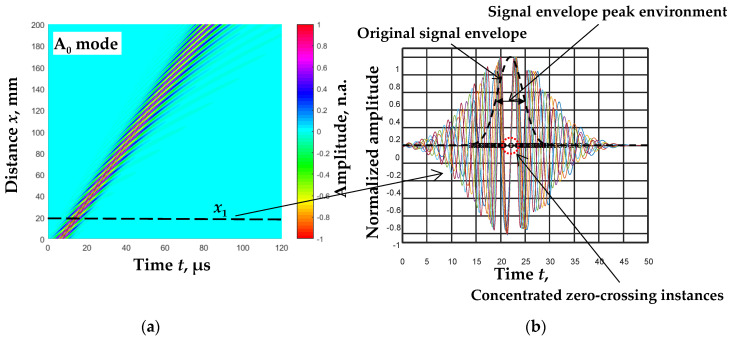
B-scan image of the simulated Lamb wave A_0_ mode in *d* = 1 mm thickness aluminum plate at *f*_ex_ = 300 kHz excitation frequency [24] (**a**) and signal filtered by five filters (color curves) at the distances *x*_1_ = 20 mm (**b**).

**Figure 2 sensors-23-06678-f002:**
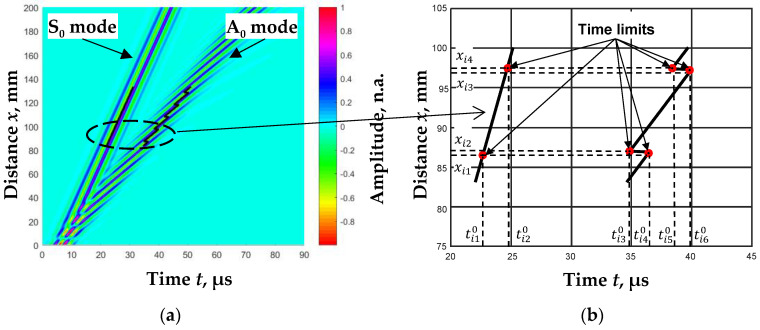
B-scan image of the simulated Lamb wave mode signals (color-coded) and calculated zero-crossing instances (line) (**a**) and zero-crossing instances of the *i*-th filter in the narrow range (**b**).

**Figure 3 sensors-23-06678-f003:**
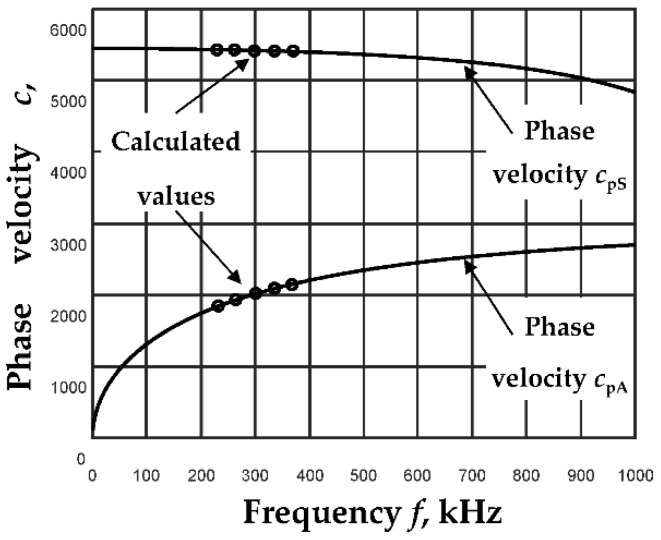
Dispersion curves were calculated using the analytical method (lines), and values of phase velocities were calculated for the proposed algorithm (dots) for *d* = 2 mm thickness aluminum plate at *f*_ex_ = 300 kHz excitation frequency.

**Figure 4 sensors-23-06678-f004:**
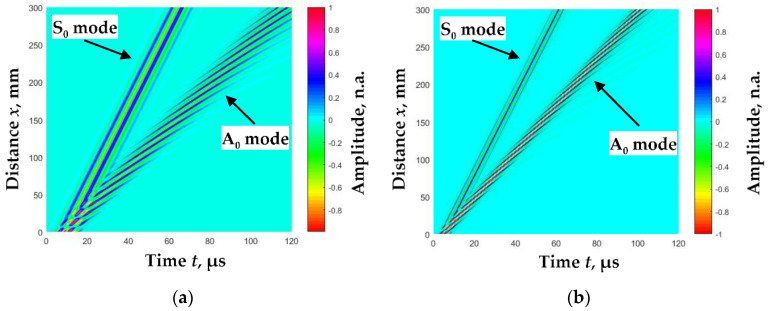
Simulated B-scan images of fundamental modes propagated in *d* = 2 mm thickness aluminum plate when: (**a**)—*f*_ex_ = 200 kHz, L_S_ = 1; (**b**)—*f*_ex_ = 400 kHz, L_S_ = 0.5.

**Figure 5 sensors-23-06678-f005:**
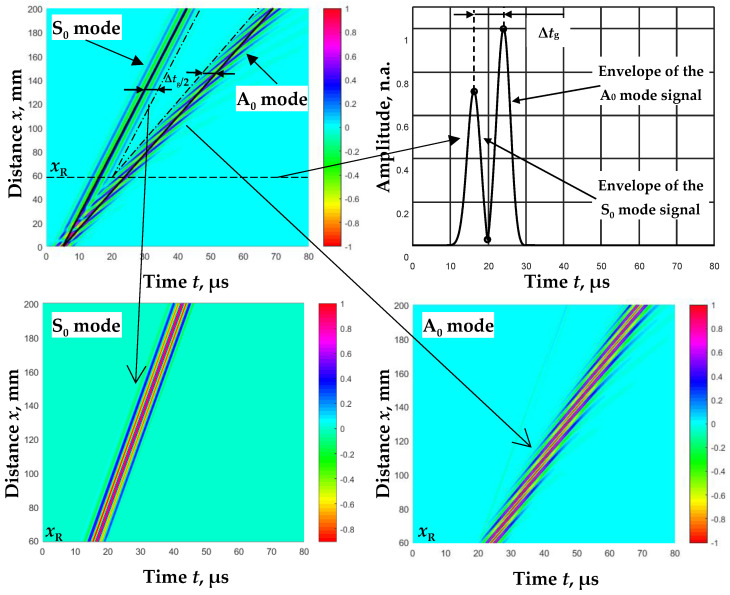
Separation of A_0_ and S_0_ modes using the signal envelopes of these modes.

**Figure 6 sensors-23-06678-f006:**
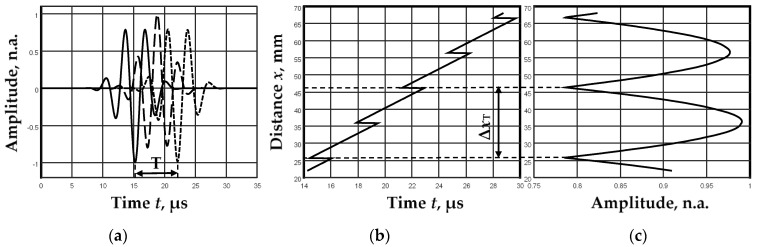
A_0_ mode signals at different distances (**a**), zero-crossing instances in the selected range of distances (**b**), and the amplitude distribution of the envelope maxima of these signals (**c**).

**Figure 7 sensors-23-06678-f007:**
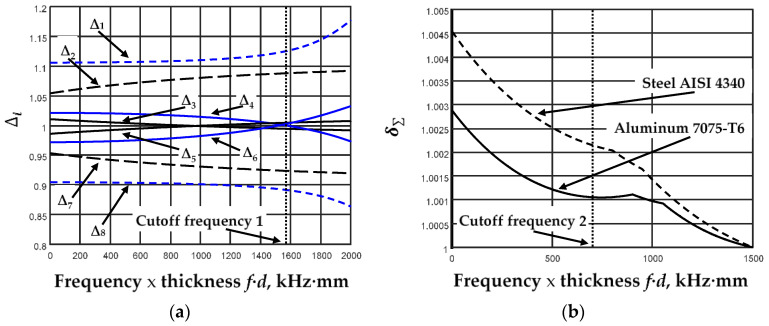
The calculated relative deviations Δ*_m_* (**a**) and normalized total relative deviation *δ*_Σ_ for aluminum 7075-T6 and steel AISI 4340 (**b**).

**Figure 8 sensors-23-06678-f008:**
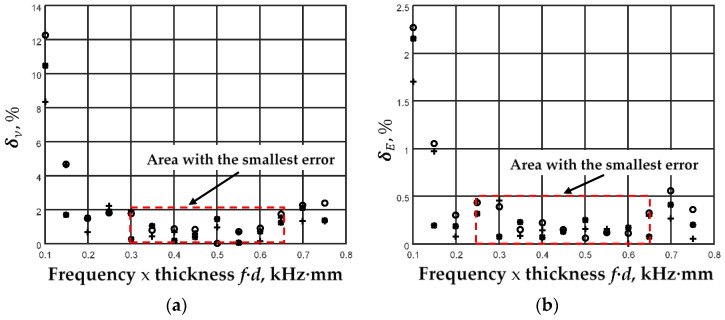
Relative error calculations result using different excitation frequencies *f*_ex_ and ratios of the amplitudes of A_0_ and S_0_ modes: ‘o’—L_S_ = 1, ‘+’—L_S_ = 0.5; ‘*’—L_S_ = 0.1. (**a**) Relative errors of Poisson’s ratio calculations; (**b**) relative errors of Young’s modulus calculations.

**Figure 9 sensors-23-06678-f009:**
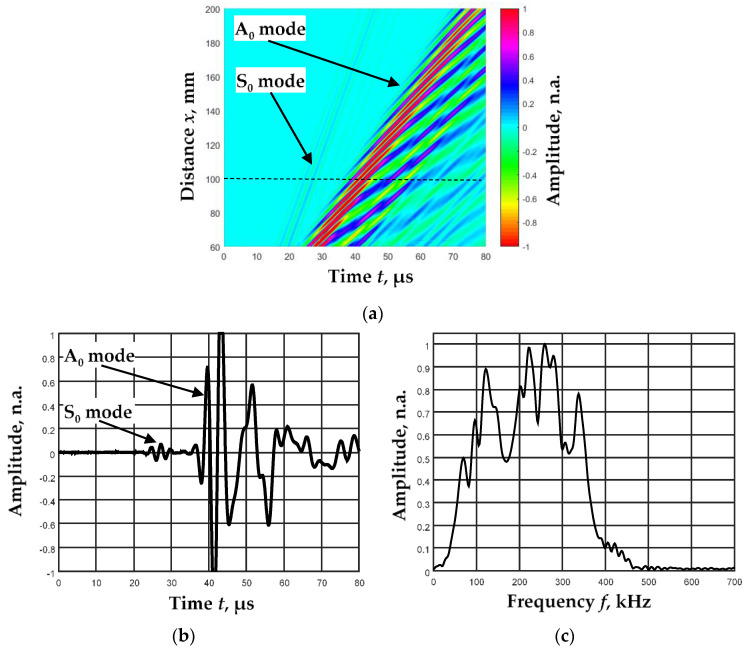
B-scan image of the Lamb wave propagating through an aluminum plate (**a**), the signal at the distance 100 mm (**b**), and the frequency response of this signal (**c**).

**Figure 10 sensors-23-06678-f010:**
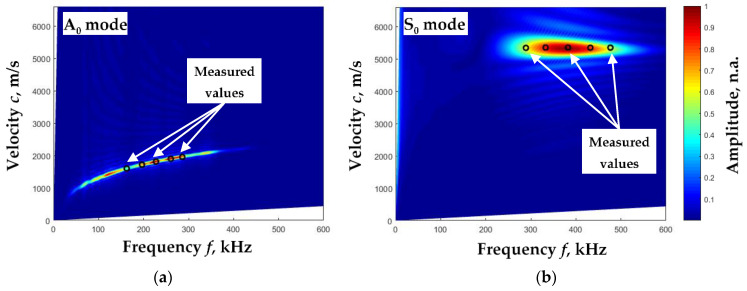
Phase velocity dispersion curves of A_0_ (**a**) and S_0_ (**b**) modes were calculated using the 2D-FFT method (color), and phase velocity values were calculated from experimental data using the proposed algorithm (dots).

**Table 1 sensors-23-06678-t001:** Parameters of phase velocity deviation calculations. The marking ◊ shows in which mode the reference values of the phase velocities are calculated.

		Δ_1_	Δ_2_	Δ_3_	Δ_4_	Δ_5_	Δ_6_	Δ_7_	Δ_8_

S_0_ mode		◊			◊		◊		◊
A_0_ mode			◊	◊		◊		◊	
Young’s modulus, *E*		+20%	+20%					−20%	−20%
Poisson’s ratio, *ν*				+20%	+20%	−20%	−20%		

## Data Availability

Not applicable.

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
