# Peer review of "Determining the Elastic Constants of Isotropic Materials by Measuring the Phase Velocities of the A0 and S0 Modes of Lamb Waves"

_sensors, 2023, doi:10.3390/s23156678_

Round 1
Reviewer 1 Report
The authors present an interesting method for determining the elastic constants of isotropic materials by measuring the phase velocities of the A0 and S0 modes of Lamb waves.
As presented, I did not see an originality of the paper about determining the elastic constants of isotropic plates. The described method pretend to solve an inverse problem using Rayleigh-Lamb equations, it is something classical, what is the added value of this paper in terms of novelty or instrumentation in NDT, for general manner? Fo example, comparing to this reference "Towards real-time assessment of anisotropic plate properties using elastic guided waves ?", N. Bochud and all, JASA 2018.
Please specify it clearly in the paper.
The english is satisfactory and the paper is well written.
Author Response
The authors present an interesting method for determining the elastic constants of isotropic materials by measuring the phase velocities of the A0 and S0 modes of Lamb waves.
As presented, I did not see an originality of the paper about determining the elastic constants of isotropic plates. The described method pretend to solve an inverse problem using Rayleigh-Lamb equations, it is something classical, what is the added value of this paper in terms of novelty or instrumentation in NDT, for general manner? For example, comparing to this reference "Towards real-time assessment of anisotropic plate properties using elastic guided waves?", N. Bochud and all, JASA 2018.
Please specify it clearly in the paper.
Thanks for the reviewer’s helpful comment. The article was supplemented with a description of the novelty of the method and its added value:
“The technique of the proposed method has several distinct advantages: l) only low-frequency fundamental Lamb wave modes are excited in the isotropic plate; 2) it was sufficient to know only the thickness and density of the plate in advance; 3) there was no need to calculate theoretical curves and compare them with experimentally obtained phase velocities values; 4) processing of ultrasonic signals was performed in real time, and values of elastic constants are obtained immediately after scanning the required distance.”
The article proposed by N. Bochud and all differs essentially in its content. It uses higher order modes excited by a linear transducer array, the elastic constants are determined using a comparison of theoretical and experimental wave numbers. It was similar in content to the article presented in [21].

Reviewer 2 Report
This manuscript describes a determination method of elastic constants of isotropic plates using Lamb wave fundamental modes. This manuscript is well written with no evident fault or mistake. However, there are no novel and original points to determine the elastic constants using a MATLAB function. This manuscript mainly focuses on the determination of the phase velocity, however, which was already proposed by the authors themselves[Ref.23]. If the merit of this research method is to obtain a very accurate result, the explanation of the research concept should be modified. At the moment, I cannot submit a positive review.
[1] Introduction, Line 131.
The authors insist that this method is simple and effective for measuring the elastic constants, but the reader cannot understand how it is simple and effective. Because if we can obtain the dispersion curve like Fig.10 by ultrasonic scan, we can estimate the phase velocity from these curves by choosing the central value. However, I think this method could be effective if this can calculate the phase velocity by a few point ultrasonic measurements.
[2] Line 187,
I can not understand why the density is easy to calculate.
[3] Table 3
It is hard to understand the calculation setup. What does the "+" mean?
In my understanding, the merit of this research method is to obtain a very accurate result. If so, what did the author aim for by choosing the twenty percent deviation?
[4] Chapter 4
The author should describe the filter in terms of using the concentrated zero-crossing instances on the time axis. Especially since this experiment uses a pulse-like signal, there seems to be significant dispersion. The reader wants to know how the author tackled them.
Author Response
This manuscript describes a determination method of elastic constants of isotropic plates using Lamb wave fundamental modes. This manuscript is well written with no evident fault or mistake. However, there are no novel and original points to determine the elastic constants using a MATLAB function. This manuscript mainly focuses on the determination of the phase velocity, however, which was already proposed by the authors themselves [Ref.23]. If the merit of this research method is to obtain a very accurate result, the explanation of the research concept should be modified. At the moment, I cannot submit a positive review.
Thanks for the reviewer’s helpful comment. The article was supplemented with a description of the novelty of the method and its added value:
“The technique of the proposed method has several distinct advantages: l) only low-frequency fundamental Lamb wave modes are excited in the isotropic plate; 2) it was sufficient to know only the thickness and density of the plate in advance; 3) there was no need to calculate theoretical curves and compare them with experimentally obtained phase velocities values; 4) processing of ultrasonic signals was performed in real time, and values of elastic constants are obtained immediately after scanning the required distance.”
The MATLAB function was only an aid to the determination of elastic constants, and the main focus of the method was the measurement of the phase velocities of the fundamental modes. The purpose of this research method was to determine elastic constants quickly, in real time and accurately.
[1] Introduction, Line 131.
The authors insist that this method is simple and effective for measuring the elastic constants, but the reader cannot understand how it is simple and effective. Because if we can obtain the dispersion curve like Fig.10 by ultrasonic scan, we can estimate the phase velocity from these curves by choosing the central value. However, I think this method could be effective if this can calculate the phase velocity by a few point ultrasonic measurements.
The advantages of the method are described in the supplement to the article above. This method does not use the 2D-FFT dispersion curve calculation method, and Fig. 10 only shows the comparison of the results of this method and the proposed method. This method can calculate the elastic constant values after scanning the required distance, which was determined by the fixed propagation jump of the A0 mode.
[2] Line 187,
I can not understand why the density is easy to calculate.
The text of the article has been corrected: “The material density ρ was either known in advance or measured by other methods”.
[3] Table 3
It is hard to understand the calculation setup. What does the "+" mean?
The + designation was replaced by the ◊ designation. The marking ◊ shows for which mode the reference values of the phase velocities are calculated.
In my understanding, the merit of this research method is to obtain a very accurate result. If so, what did the author aim for by choosing the twenty percent deviation?
Large deviations of the elastic constants (±20%) were chosen in order to find out how much such deviations influence the deviations of the phase velocities from the reference values.
[4] Chapter 4
The author should describe the filter in terms of using the concentrated zero-crossing instances on the time axis. Especially since this experiment uses a pulse-like signal, there seems to be significant dispersion. The reader wants to know how the author tackled them.
Thanks for the very good comment. Information about the filters used during processing of experimental results was provided in the supplementary text of the article.
“Filter packets with different center frequencies are selected for these modes according to the algorithm described in detail [24]. Five filter (N=5) packets have been selected for both modes. The following filter parameters have been selected for A0 mode: resonant frequency of the center filter f3=226.8 kHz, bandwidths of the filters Δf=46.4 kHz, the distances between the filters df=34.8 kHz. Parameters corresponding to S0 mode: f3=382.1 kHz, Δf=75 kHz, df=56.2 kHz.”

Round 2
Reviewer 1 Report
The paper can be now published as it is
Reviewer 2 Report
I confirmed the points which the reviewer requested were modified or added.